# Chronic Lymphocytic Leukemia: Management of Adverse Events in the Era of Targeted Agents

**DOI:** 10.3390/cancers16111996

**Published:** 2024-05-24

**Authors:** Andrea Galitzia, Monica Maccaferri, Francesca Romana Mauro, Roberta Murru, Roberto Marasca

**Affiliations:** 1Hematology and Stem Cell Transplantation Unit, Ospedale San Francesco, 08100 Nuoro, Italy; andrea.galitzia@aslnuoro.it; 2Hematology Unit, Department of Oncology and Hematology, A.O.U of Modena, Policlinico, 41125 Modena, Italy; maccaferri.monica@aou.mo.it (M.M.); roberto.marasca@unimore.it (R.M.); 3Hematology, Department of Translational and Precision Medicine, Sapienza University, 00185 Rome, Italy; mauro@bce.uniroma1.it; 4Hematology and Stem Cell Transplantation Unit, Ospedale Oncologico A. Businco, ARNAS G. Brotzu, 09134 Cagliari, Italy; 5Department of Medical and Surgical Sciences, Section of Hematology, University of Modena and Reggio Emilia, 41121 Modena, Italy

**Keywords:** Chronic Lymphocytic Leukemia (CLL), targeted therapies, Bruton Tyrosine Kinase inhibitors (BTKis), BCL-2 inhibitors (BCL-2is), adverse event management, Tumor Lysis Syndrome (TLS), cardiac toxicity, hematological toxicity, drug–drug interactions, infection management

## Abstract

**Simple Summary:**

Targeted agents, such as Bruton’s Tyrosine Kinase inhibitors and BCL-2 inhibitors, have transformed the treatment landscape for Chronic Lymphocytic Leukemia (CLL). While these therapies offer a more precise approach compared to traditional chemotherapy, they also bring a spectrum of adverse events that can impact patient health and treatment efficacy. This review aims to provide a comprehensive overview of the management strategies for these AEs, encompassing cardiac complications, Tumor Lysis Syndrome, bleeding risks, and other organ-specific challenges. Our goal is to equip healthcare professionals with the knowledge to effectively manage these adverse events, enhancing patient safety and optimizing therapeutic outcomes. The insights from this review will contribute to better clinical decision-making and patient care in CLL treatment, highlighting the significance of personalized management in the era of targeted therapy.

**Abstract:**

The treatment landscape for CLL has undergone a profound transformation with the advent of targeted agents (TAs) like Bruton’s Tyrosine Kinase inhibitors (BTKis) and BCL-2 inhibitors (BCL-2is). These agents target crucial cellular pathways in CLL, offering superior efficacy over traditional chemo-immunotherapy, which has led to improved progression-free and overall survival rates. This advancement promises enhanced disease control and potentially normal life expectancy for many patients. However, the journey is not without challenges, as these TAs are associated with a range of adverse events (AEs) that can impact treatment efficacy and patient quality of life. This review focuses on detailing the various AEs related to TA management in CLL, evaluating their frequency and clinical impact. The aim is to present a comprehensive guide to the effective management of these AEs, ensuring optimal tolerability and efficacy of TAs. By reviewing the existing literature and consolidating findings, we provide insights into AE management, which is crucial for maximizing patient outcomes in CLL therapy.

## 1. Introduction to Toxicity Profile of Targeted Therapies and Management of Intolerance

Targeted agents (TAs) have dramatically changed the treatment of Chronic Lymphocytic Leukemia/Small Lymphocytic Lymphoma (CLL/SLL) in the last decade, leading to the shift from chemo-immunotherapy to TA-based therapy [1,2].

The small molecule therapies, Bruton Tyrosine Kinase inhibitor (BTKi) and B-cell Lymphoma 2 inhibitor (BCL-2i) target specific and relevant cellular pathways critical to CLL biology which sustain cellular proliferation and survival. TAs have demonstrated superior efficacy with respect to chemo-immunotherapy, in terms of progression-free survival (PFS) and overall survival (OS), in both randomized trials and real-life studies [3,4,5,6,7,8].

BTKi, used either as a single agent or in combination with anti-CD20 monoclonal antibodies (MoAb), is characterized by a high overall response rate (ORR), prolonged PFS and OS, and a low rate of complete remission (CR). This necessitates continuous, indefinite treatment until intolerance or disease progression becomes evident. Conversely, venetoclax-based treatments are characterized by a higher rate of CR, especially when associated with CD20 MoAb, achieving deep responses and providing a high rate of undetectable minimal residual disease, and are administered as predefined, fixed-duration treatments [9,10,11]. Despite the impressive efficacy of TA, it is crucial to acknowledge the significant AE profile associated with these drugs. The adverse events (AEs) of TAs typically manifest as class effects, such as cardiological toxicity with BTKi and tumor lysis syndrome (TLS) with BCL-2i. Therefore, when identifying treatment, selected clinical factors must be thoroughly evaluated. These include pre-existing comorbidities, such as severe cardiovascular conditions, a history of bleeding episodes, or the need for permanent anticoagulation (particularly relevant for BTKi), as well as renal dysfunction (a consideration with venetoclax). This pre-treatment evaluation should be extended to concomitant medications that may exacerbate TA-related AEs, like Cytochrome P450 3A (CYP3A) inhibitors and some classes of anticoagulants (Figure 1) [1,2,12].

Management of AEs extends beyond dose adjustments or treatment discontinuation and includes vigilant monitoring and targeted interventions for specific complications. In this context, perfect collaboration between specialists, employing a multidisciplinary approach, leads to achieving better outcomes [13]. Decisions regarding treatment discontinuation or switching to alternatives with more favorable safety profiles should be made judiciously, balancing the risk of disease progression against the potential for reducing adverse effects and thus maximizing patient outcomes.

Awareness and optimal management of these adverse events are essential to minimize treatment discontinuations, maximize efficacy, enhance survival rates, and preserve a good quality of life for CLL patients. In the following section, the key AEs of TAs and measures for their management are summarized. 

## 2. Class Effect Profile and Adverse Events Management: BTK Inhibitors

The FDA and EMA have approved several BTKis for the treatment of CLL/SLL, including the first-generation ibrutinib and the second-generation inhibitors acalabrutinib and zanubrutinib. Pirtobrutinib, a non-covalent BTKi, retains kinase inhibition even in the presence of a BTK C481 mutation [14] and demonstrates high specificity for BTK, with minimal off-target effects [15]. Pirtobrutinib addresses the unmet need for alternative therapies in patients who are previously treated, resistant, and/or intolerant to irreversible BTKi, showing a favorable side-effect profile [15,16].

Other covalent BTKis, including spebrutinib and tirabrutinib, have been investigated for potential applications in specific lymphomas but are not yet FDA-approved [6].

The toxicity profiles of BTKis are closely linked to their kinase-binding patterns, including both on-target inhibition of BTK and variable off-target inhibition of other kinases, such as interleukin-2-inducible T-cell kinase (ITK), tyrosine kinase expressed in hepatocellular carcinoma (TEC), and epidermal growth factor receptor (EGFR) family kinases. Ibrutinib, notably, targets additional kinases such as C-terminal Src kinase (CSK), bone-marrow tyrosine kinase gene on chromosome X (BMX), B lymphocyte kinase (BLK), and Janus kinase 3 (JAK3) [17].

The long-term toxicity profile of ibrutinib is well-defined, with over eight years of follow-up data available since its initial pivotal study [7]. AEs are the primary cause of ibrutinib discontinuation and include heart arrhythmias, bleeding, diarrhea, arthralgias, hypertension, and infections [18,19]. Acalabrutinib and zanubrutinib are more selective next-generation covalent BTKis and exhibit less off-target activity and better tolerability compared to ibrutinib [20,21,22,23,24].

### 2.1. Cardiotoxicity

The use of BTKis has been associated with a significant risk of cardiovascular AEs, including atrial fibrillation/flutter (AF), hypertension, heart failure (HF), sudden cardiac death (SCD), and ventricular arrhythmias (VA) [25,26].

The emergence of these events poses a significant challenge for the continuation of this crucial therapy. A pragmatic, case-specific approach involving cardio-oncologists is recommended. Looking ahead, high-quality trials are needed to establish evidence-based guidance for clinicians regarding optimal management strategies in this population [27].

A summary of cardiac AEs associated with BTKis’ supposed mechanism and proposed management strategy is provided in Table 1.

#### 2.1.1. Atrial Fibrillation

Cancer is ‘per se’ associated with an increased risk of AF, with patients diagnosed with tumors showing a 20% higher adjusted risk of AF compared to those without cancer, irrespective of ongoing treatment [25].

Concerning hematological malignancies, the likelihood of AF seems notably elevated in individuals with CLL. In a Mayo Clinic study, approximately 6% of CLL patients had a prior history of AF. Around 6.1% of patients without a history of AF developed incident AF during the follow-up period, resulting in an annual incidence of about 1% [28]. Moreover, over a 10-year period, the incidence of AF in patients treated with ibrutinib was highly variable, with rates ranging from 4% to 33% of patients depending on the combination of risk factors [28]. The RESONATE trial reported a 5% incidence of AF [29,30]. Among patients aged ≥ 65 years, the cumulative incidence of AF over 60 months reached a 16% rate, and up to 11% of patients experienced severe events [31]. A pooled analysis of four randomized controlled trials involving patients with CLL and mantle cell lymphoma (MCL) treated with ibrutinib found AF rates of 6.5% at 16.6 months and 10.4% at 36 months [32,33]. 

While AF events were more frequently observed with ibrutinib, they have been recorded with any BTKi currently approved for use in clinical practice. This observation defines AF as a specific class effect of BTKi [34].

Patients receiving next-generation BTKis, such as acalabrutinib and zanubrutinib, exhibited fewer cardiac adverse effects compared to those treated with ibrutinib [35,36]. A pooled analysis of clinical trials investigating acalabrutinib showed an AF incidence ranging from 5% to 9% [37]. In the ELEVATE-RR trial, a lower incidence of AF was reported with acalabrutinib than ibrutinib (9% vs. 16%) [38]. Similarly, a phase III trial comparing zanubrutinib to ibrutinib in relapsed/refractory (RR)-CLL patients also reported a lower incidence of AF in zanubrutinib-treated patients (5.2 vs. 13.3%) [35]. Despite the short follow-up of the Bruin study investigating pirtobrutinib, preliminary data indicate that the rate of AF events is lower than 5% [15,16].

AF induced by BTKis is more prone to occur early in the treatment, with the risk of severe events decreasing over time [37,39]. Risk factors associated with incident AF include age, male gender, valvular heart disease, and hypertension. The prevalence of AF exhibited an age-dependent pattern among both CLL patients and age-matched adult control populations. In CLL patients, a higher prevalence was observed across all age groups [28]. Additional risk factors include pre-existing heart failure and left atrial abnormality on electrocardiogram [40]. These factors can be applied for risk stratification, and, consequently, scoring systems have been developed to identify patients at elevated risk who could benefit from increased monitoring to minimize complications [41,42].

In the RESONATE trial, AF showed no impact on long-term survival outcomes [29,30]. However, other findings suggested that a new onset of AF events was associated with a higher risk of thromboembolism, stroke, and heart failure, leading to ibrutinib discontinuation [43,44]. Moreover, patients who experience AF while on a BTKi showed poorer long-term outcomes than those without AF with an inferior PFS and an around threefold rise in long-term mortality [37]. The unfavorable outcome of these patients might be linked to treatment interruptions or bleeding and stroke events. Conversely, it is still unclear whether adjustments in dosage or changes in therapy contribute to the worse outcome.

##### Mechanisms Leading to BTKi-Induced AF

The mechanisms underlying AF events associated with BTKi remain not completely understood. Experimental data suggest the involvement of both potential on- and off-target pathways. In particular, the inhibition of CSK is considered a potential factor favoring the onset of AF [45]. Other mechanisms proposed to explain BTKi-induced arrhythmias, particularly with ibrutinib, suggest the inhibition of the PI3K/AKT and ion channel pathways. In particular, data from in vitro studies indicate that ibrutinib may hinder various currents of the cardiac action potential through direct binding of this agent to ion channels [38,46]. Moreover, cardiac remodeling and fibrosis may also play a role in the downstream manifestation of AF due to ibrutinib [46].

##### Management of AF in Patients Receiving BTKi

The management of AF associated with BTKi treatment is challenging due to the increased bleeding risk associated with anticoagulation and the inherent bleeding risks associated with BTK inhibition [47]. Moreover, drug–drug interactions (DDI) with standard antiarrhythmics should be considered. Before initiating BTKi treatment a complete cardiac assessment is strongly recommended, including a cardiovascular history, blood pressure measurement, and a 12-lead ECG [48]. For patients with more risk factors for cardiovascular events (e.g., age over 50 years, prior cardiac arrhythmia, hypertension, heart failure, valvular disease, or prior administration of cardiotoxic therapy), a baseline echocardiogram is also advised [48,49]. In patients with well-controlled pre-existing AF, as well as those deemed high-risk by cardiac risk assessment, initiation of BTKis may be safely considered. In these cases, second-generation BTKis are preferred over ibrutinib [50]. Close clinical monitoring of patients receiving BTKis, especially during the initial six to eight months of therapy, is essential to identify cases of asymptomatic paroxysmal AF. 

The first treatment approach in managing AF includes appropriate anticoagulation and the evaluation of the optimal treatment required to restore sinus rhythm or obtain rhythm control [48,51].

Essa et al. proposed the Atrial Fibrillation Better Care (ABC) pathway as a comprehensive management strategy for AF patients, focusing particularly on factors favoring rhythm control, such as younger patients without structural heart disease; hemodynamic instability; persistent symptoms despite medical management; AF secondary to correctable causes [27].

The primary objectives of rate control in patients with acute or chronic AF with a rapid ventricular response are to alleviate symptoms and to minimize the risk of developing LV systolic dysfunction. Typically, non-dihydropyridine calcium channel blockers (diltiazem and verapamil) and beta-blockers are employed to manage the heart rate [51]. For patients treated with BTKi, beta-blockers are generally preferred over calcium channel blockers due to potential drug interactions with BTKis, which are mediated through CYP3A4 inhibition. Additionally, digoxin should be avoided in combination with ibrutinib because of interactions via P-gp [27].

When managing AF, it is crucial to assess the risk of bleeding and stroke. Clinical tools like the CHA2DS2-VASc and HAS-BLED scores are valuable for evaluating these risks and guiding treatment decisions. Current guidelines recommend anticoagulation therapy for patients with AF who have an estimated annual thromboembolic risk of 2% or higher, indicated by a CHA2DS2-VASc score ≥ 2 in men and ≥3 in women, to prevent stroke and systemic thromboembolism [51]. Direct oral anticoagulants (DOACs) are usually the preferred choice for anticoagulation, offering simpler administration, more consistent pharmacologic effects, and fewer DDIs. Nevertheless, anticoagulants further increase the intrinsic bleeding risk associated with BTKis [30,37,52]. For patients with a CHA2DS2-VASc score of 1 (or 2 in women), anticoagulation is also reasonable; however, the decision to treat with anticoagulants may sometimes require additional discussion with patients, as the strength of the recommendation is less robust [51]. Therefore, if these patients have additional risk factors for bleeding (e.g., baseline thrombocytopenia, high HAS-BLED score) they should not receive anticoagulation [48,53]. In patients with CHA2DS2-VASc scores < 2, continuing ibrutinib has been recommended with or without anticoagulation based on their bleeding risk profile [50,54]. 

The choice of DOAC should be based on patient-specific factors and shared decision-making [55]. Generally, Xa inhibitors should be preferred over dabigatran due to the necessary modifications in cases of impaired renal function, and fewer DDIs. Daily DOACs, such as rivaroxaban or edoxaban, may be considered for patients facing challenges with twice-daily administration. Apixaban is preferred by some clinicians due to retrospective and indirect comparisons indicating more favorable efficacy and safety, and the relatively lower dependence on renal clearance [56,57].

Warfarin has not been tested for concurrent use with BTKis and is not recommended due to the increased risk of bleeding [58]. In patients who need anticoagulation with warfarin, an alternative treatment for CLL should be considered.

Lastly, there are scarce data regarding the effectiveness of rhythm control measures in patients treated with BTKis, such as cardioversion or ablation, and the decision to explore these options should be carefully evaluated on a case-by-case basis according to cardiological indications [51]. This approach might prove beneficial in alleviating the antiarrhythmic therapy burden for individuals experiencing paroxysmal AF events while on a BTKi [58].

If an AF event occurs during BTKi treatment, according to EMA and FDA recommendations, an in-depth risk–benefit assessment is required to evaluate the risk and benefit of maintaining, or not, BTKi treatment. If disease progression occurs after BTKi cessation due to severe cardiac toxicity, alternative treatment options, such as Bcl-2 antagonists, are recommended given their overall safer cardiac profile [58].

A collaborative and multidisciplinary approach to AF management is essential, necessitating the active participation of both the treating hematologist and cardiologist. Moreover, we emphasized how optimizing the safety and long-term tolerability of BTKis requires a thorough evaluation of each patient’s clinical history before initiating therapy. Additionally, careful attention should be given to patient-reported signs and symptoms observed during the course of treatment [13,50,59].

#### 2.1.2. Ventricular Arrhythmia

In recent years, there has been increasing evidence of the risk of VA and SCD in patients undergoing BTKi treatment.

Lampson et al. identified a total of 10 cases of SCD or cardiac arrest among ~1000 CLL patients treated with ibrutinib in clinical trials. They estimated an incidence rate of 788 events per 100,000 person-years, which is significantly higher than what is expected in the general population of the same age group [60].

Salem et al. conducted an extensive analysis of pharmacovigilance datasets, identifying a higher frequency of reports on VAs associated with ibrutinib compared to other drugs. These arrhythmias occurred early after drug initiation, with a median time of 70 days [26]. Further investigations revealed that ibrutinib poses a distinct risk for VAs and SCD, which is independent of any changes in QT duration [61]. 

VAs may be a class effect of BTKi therapy. A recent study reported that patients treated with acalabrutinib have a more than eightfold risk of VAs and SCD events [62]. Regarding zanubrutinib, recent data from the SEQUOIA trial indicate that zanubrutinib-treated patients showed a notable risk of VAs [52].

Indeed, while additional data is awaited, the documented occurrences of VAs and SCD associated with ibrutinib and acalabrutinib do not seem to impact the treatment’s efficacy in improving survival for CLL, when compared to other therapies. A comprehensive evaluation awaits long-term data on zanubrutinib and for all BTKis [34].

The factors contributing to VAs associated with BTKis remain largely unidentified. The mechanisms underlying VAs induced by BTKis are still not fully understood. However, disruptions in calcium handling and myocardial fibrosis, altered repolarization dynamics, and direct effects on cardiac ion channels have been described as associated with BTKi treatment, leading to an increased risk of arrhythmic ventricular events [63,64]. Existing data indicate that age may increase the associated risk [62]. Furthermore, individuals with a prior history of symptomatic VAs, uncontrolled or persistent heart failure, or familial instances of sudden cardiac death are strongly advised to consider alternative therapy. Additionally, regular interim ECG assessments, at least quarterly during the initial 12 to 18 months, may be useful [58]. 

For patients experiencing symptomatic VAs during treatment, some authors suggest a hold or discontinuation of therapy followed by an additional cardiological assessment to prognosticate the risk of recurrent cardiac events [58]. However, given their potential life-threatening consequences, definitive discontinuation of BTKis is recommended, with alternative treatment in the case of disease progression. 

#### 2.1.3. Hypertension

All approved covalent BTKis are known to be associated with hypertension, which can develop at any point during treatment.

In a comprehensive meta-analysis of clinical trials involving ibrutinib, exposure to the drug was associated with a threefold increase in risk for the onset of hypertension [65]. A single-center study reported that 78.3% of patients treated with ibrutinib experienced either new-onset or worsened hypertension, with a median time of development of 6 months [66].

In a recent assessment of patients treated with acalabrutinib, comparable though less pronounced increases in blood pressure values were noted [38,67].

Finally, patients treated with zanubrutinib had similar rates of significant hypertension to patients on ibrutinib [52].

Like BTKi-related AF, the inhibition of the Phosphoinositide 3-kinase (PI3K) pathway may play a role in cellular remodeling mechanisms, potentially leading to vascular tissue fibrosis; however, the cause of hypertension observed in BTKi treatment remains uncertain, as it is unclear whether it results from off-target kinase inhibition or systemic inflammatory changes that increase the risk of hypertension development [58].

Before initiating treatment with BTKis, optimal baseline management of hypertension is required. This does not preclude the inclusion of patients with a history of well-controlled hypertension, provided that closer monitoring and a thorough pre-treatment assessment are conducted to ascertain the risk of worsening. Additionally, it is crucial in clinical practice to address predisposing factors, such as stress, pain, excessive alcohol consumption, renal impairment, untreated sleep apnea, obesity, and a sedentary lifestyle [48].

Blood pressure should be monitored twice a week during BTKi treatment, and consistent communication is recommended to identify any deviations from the pretreatment baseline. In particular, if patients develop hypertension during BTKi treatment, early intervention with guideline-directed antihypertensive medication remains the standard of care [58]. In cases where blood pressure increase is significant, combinations of antihypertensive agents may be required to achieve a blood pressure target, in line with currently published guidelines [48].

If grade ≥3 hypertension occurs, BTKi interruption may be considered as a prudent initial measure, with potential dose reduction upon achieving a reasonably controlled blood pressure. In cases of recurrent severe episodes during ibrutinib therapy, transitioning to an alternative BTKi like acalabrutinib is reasonable, acknowledging the possibility of residual hypertension risk [58].

#### 2.1.4. Heart Failure

Although no significant signal for HF was observed in the initial clinical trials, recent findings from the aggregated long-term follow-up of later-phase ibrutinib trials indicate a potential rise in the risk of HF, with up to 5% of patients experiencing HF often manifesting years after starting treatment [7,68].

The potential development of HF can be influenced by the additional cardiotoxic effects of BTKis. Furthermore, current evidence indicates that the presence of preexisting AF may increase the risk of HF, particularly in patients using ibrutinib. Similarly, a significant proportion of individuals on ibrutinib may show signs of myocardial fibrosis on cardiovascular magnetic resonance (CMR) imaging [69].

Data regarding the safety of implementing a standardized cutoff in baseline LV ejection fraction for initiating BTKis are lacking, as most trials excluded patients with known baseline heart failure. For patients with a history of previous heart failure, we prefer next-generation BTKis to reduce the risk of additional cardiotoxic effects, such as AF or uncontrolled hypertension. These conditions are often poorly tolerated in these patients and may contribute to further decompensation [58]. Considering alternative therapies is warranted, based on clinical indications, for effective CLL control.

A summary of the general management of BTKis in case of heart failure and severe cardiac arrhythmias is provided in Figure 2.

#### 2.1.5. Stroke

Given the established association with AF, it is reasonable to infer that BTKis may also elevate the risk of stroke. A recent SEER-Medicare study involving older patients correlated the use of ibrutinib to an almost twofold rise in the risk of stroke attributable to AF [76]. 

For the assessment of a patient’s stroke risk and management, refer to the “Management of AF in patients receiving BTKi” section. Additionally, it is crucial to proactively address and optimize other factors contributing to stroke risk, such as hypertension, diabetes, and hyperlipidemia, in all patients receiving BTKis [58].

#### 2.1.6. Other Cardiac Complications

The use of ibrutinib has been linked to reversible non-ischemic cardiomyopathy. Symptoms such as systolic dysfunction, decline in left ventricular ejection fraction, and reversible heart failure are likely indicators [69]. Takotsubo cardiomyopathy, also known as apical ballooning syndrome or stress cardiomyopathy [77], was a reported adverse effect in patients treated with tyrosine kinase inhibitors, and, some years ago, a case report on mid-cavitary Takotsubo cardiomyopathy post-ibrutinib therapy was described [78].

### 2.2. Bleeding

Bleeding events are common in patients treated with BTKis, though major events are rare. In patients treated with ibrutinib, any-grade bleeding events have been reported in up to 67% with grade ≥ 3 events observed in from 2 to 9% of cases [79]. For acalabrutinib, any-grade bleeding has been reported in from 18% to 66% of patients, and, with grade ≥3 events, from 0% to 3% [20,36,80]. Among patients treated with zanubrutinib, from 44% to 51% of patients experienced any grade bleeding events with from 3% to 5% having major bleeding events [35]. In the ELEVATE-RR trial, a higher rate of bleeding events was observed in ibrutinib-treated patients compared to those treated with acalabrutinib [81].

Bleeding events related to BTKis are due to the off-target inhibition of Tec kinase, which disrupts platelet activation pathways and suppresses Glycoprotein VI (GPVI) signaling [82,83,84,85]. The concurrent administration of antiplatelet drugs, such as aspirin or P2Y12 inhibitors, with BTKis further increases the risk of bleeding. Therefore, it should be avoided, and alternative treatment approaches should be considered [86,87].

Risk factors for severe bleeding in patients treated with ibrutinib include the concomitant use of anticoagulants or antiplatelet agents, baseline thrombocytopenia, and an elevated HAS-BLED score [88]. Considering these factors, along with other pre-existing conditions that may increase the bleeding risk (e.g., hypertension or liver disease), is essential before initiating BTKi treatment. Although the HAS-BLED scale is not specific to BTKi treatment, it can assist in this evaluation [89,90].

Most bleeding events are mild and do not require dose adjustments or discontinuation. In a retrospective analysis of the Mayo Clinic, 91/209 CLL patients required a dose interruption, and only 5 were due to bleeding [91]. Moreover, the incidence of minor bleeding is not associated with an increased risk of major hemorrhage and diminishes over time, reaching a plateau within the first six months [88].

As discussed in the previous sections, 10–12% of patients develop AF and necessitate anticoagulation, thereby elevating the risk of grade 3–4 bleeding. These patients need close monitoring for signs of bleeding. 

In patients requiring planned surgery, it is recommended to discontinue BTKis for from 3 to 7 days both before and after surgery, with the duration dependent on the type of surgery [19]. Nevertheless, the given risk and benefit of holding BTKi treatment for less urgent procedures should be discussed with the patient depending on their disease control and clinical status [58]. 

Because BTKis induce an irreversible inhibition of platelet aggregation, patients with major hemorrhages require platelet transfusion, regardless of platelet count. In the case of a life-threatening hemorrhage, BTKis should be definitively discontinued.

### 2.3. Hematological Toxicity

Grade ≥ 3 neutropenia has been recorded in BTKi-treated patients as an on-target toxicity effect, in particular, in 10% of patients treated with ibrutinib, in up to 44% of those treated with zanubrutinib [92,93,94], in 10–16% of those treated with acalabrutinib monotherapy, and in up to 30% of those treated with acalabrutinib combined with obinutuzumab [22,95]. Usually, neutropenia improves with growth factor support and does not require dose interruption or discontinuation [13,50,96]. 

Thrombocytopenia in CLL patients can stem from various factors, such as splenomegaly, bone marrow failure due to tumor infiltration, previous chemotherapy, or megakaryocyte dysplasia. Acalabrutinib monotherapy has been associated with thrombocytopenia in 7–11% of patients, with grade ≥ 3 events reported in 3–4% of cases [22,97]. In treatment-naïve (TN) patients receiving zanubrutinib, 6% experienced thrombocytopenia grade ≥ 3 [98], whereas, in previously treated patients, thrombocytopenia of all grades and grade ≥ 3 occurred in 42% and 15%, respectively [94].

Dose modifications, or withholding treatments, are strongly recommended in patients with severe cytopenias treated with all BTKis, with dosage adjustments indicated by the drug labels [70,71,72,73,74,75]. 

To rule out other possible causes of cytopenias, such as myelodysplasia or persistent bone marrow infiltration, bone marrow evaluation should be considered [99].

A summary of hematological toxicities along with other AEs associated with BTKis, including their supposed mechanism and proposed management strategy, is provided in Table 2.

### 2.4. Gastrointestinal Events

Diarrhea is a frequent AE observed with BTKi treatment, typically appearing early within the first six months and usually self-limiting. The incidence of diarrhea in ibrutinib-treated patients ranges from 20% to 50%, depending on the duration of the follow-up and prior treatment [24,31,100,101]. Grade 3–4 events are relatively rare, occurring in from 1% to 6% of cases [68,101]. Comparative studies show that ibrutinib has higher rates of gastrointestinal side effects than acalabrutinib and zanubrutinib [24,38]. However, low-grade diarrhea is a common AE even for second-generation BTKis despite their higher selectivity, with incidences of 20–35% and 13–37.5%, for acalabrutinib and zanubrutinib, respectively [24,38,52,80,102,103]. Gastrointestinal toxicities associated with BTKis are likely due to off-target effects on EGFR, which plays a crucial role in maintaining the normal function of the gastrointestinal tract. On-target effects may also impact gut immune homeostasis [99,104,105].

Clinical management primarily involves symptomatic treatment and dose adjustments. Mild to moderate cases often respond to dietary modifications, hydration, and anti-diarrheal medications like loperamide. Severe or persistent cases may necessitate dose reduction or temporary discontinuation of BTKi therapy. Probiotics might be beneficial in restoring gut flora [50]. For patients intolerant to initial treatment, switching to an alternative BTKi with a more favorable gastrointestinal profile, such as acalabrutinib or zanubrutinib, has shown improved tolerability [106,107]. In complex cases, consulting a gastroenterologist can provide additional insights.

### 2.5. Dermatological Complications

Mild to moderate dermatological toxicities have been observed in 2–27% of patients receiving ibrutinib within the first year of treatment [108]. Acalabrutinib and zanubrutinib also exhibit dermatological toxic effects comparable to those reported with ibrutinib.

Dermatologic AEs induced by BTKis have been considered to result from an off-target effect on the EGFR and similar toxicities are also described in patients treated with EGFR inhibitors [109].

Rash stands out as the prevailing cutaneous side effect, with two distinct subtypes: a non-palpable, mostly asymptomatic petechial rash, and a non-blanching, violaceous papular pruritic eruption closely resembling leukocytoclastic vasculitis [110]. Aphthous-like ulcerations with stomatitis, neutrophilic dermatosis, and skin cracking can also occur.

Patients respond well to topical corticosteroid therapy. In cases of severe rash, oral antihistamines and systemic corticosteroids are employed, and a temporary interruption of ibrutinib may prove beneficial [111,112].

Nail and hair changes are also common, being documented in 26% and 66% of patients, respectively. Hair alterations manifest as softening and straightening, while nail changes present as brittle fingernails or splitting. Typically, these changes become apparent around six months after initiating ibrutinib treatment, aligning with the natural growth cycle of nails. Supplementation with biotin may prove beneficial [111]. 

### 2.6. Arthralgias and Myalgias 

Arthralgias and myalgias are frequently observed in CLL patients receiving ibrutinib treatment with rates ranging between 11% and 36% [113,114]. Risk factors include younger age, female sex, and ibrutinib use as the first treatment [113]. Usually, most patients experience low-grade events, with a minority discontinuing therapy for grade ≥ 3 events [54].

Symptoms may be relieved by taking NSAIDs, acetaminophen, or corticosteroids; caution should be exercised when using these medications due to the potential risk of bleeding. 

Mechanisms underlying ibrutinib-induced arthralgias/myalgias are unclear but are likely related to off-target inhibition [113]

When symptoms impact daily activities, dose reduction, and, in the absence of improvement, temporary discontinuation of ibrutinib is recommended [113]. If symptoms recur upon rechallenge, alternative treatment should be considered. Recent data suggest that approximately two-thirds of patients with ibrutinib-induced arthralgias/myalgias did not experience recurring symptoms when treated with acalabrutinib [95]. Zanubrutinib is also a reasonable choice in patients intolerant to ibrutinib or acalabrutinib [107].

### 2.7. Headaches

Acalabrutinib is associated with a distinct manifestation of headaches. In pivotal studies, approximately 70% of patients experienced grade 1–2 headaches, predominantly occurring in the first weeks of treatment and enduring for a limited time period [21,102,115]. Typically, these headaches arise within 30 min of dosing, often requiring no medical intervention. Treatment discontinuation due to headache has been reported in only 1% of cases [80]. Use of a moderate dose of caffeine or acetaminophen is admitted, while it is advisable to avoid the use of nonsteroidal anti-inflammatory drugs (NSAIDs), if possible [116,117]. The exact mechanisms remain unclear, and the potential involvement of calcitonin gene-related peptide (CGRP) agonism has also been considered. This mechanism is particularly noteworthy considering the emergence of a new class of migraine medications designed to counteract CGRP [118].

## 3. Class Effect Profile and Adverse Events Management: BCL-2 Inhibitors

Almost all CLL cells evade apoptosis by overexpression of the BCL-2 protein. The BCL-2 protein is highly expressed in CLL, where it sequesters and blocks the function of the Bcl-2 Homology 3 domain (BH3)-only pro-apoptotic proteins, conferring resistance to apoptosis. Inhibition of this pathway constitutes a very attractive and powerful therapeutic target [119]. Venetoclax (ABT199), a highly selective BCL-2i targeting specifically the BH3, induces a rapid reduction of the CLL disease burden. It displaces BCL2-family pro-apoptotic proteins, leading to mitochondrial outer membrane permeabilization, activation of caspases, and subsequent cell apoptosis [120,121].

Venetoclax is administered orally once daily at a full dose of 400 mg/d. However, the starting dose is 20 mg once daily for 7 days, which is gradually increased over 5 weeks to the full daily dose. This 5-week dose ramping schedule was designed to gradually reduce tumor burden and decrease the risk of TLS [122]. Food increases the absorption of venetoclax, so it is recommended to take it once daily, preferentially with meals containing fat. At least during the ramp-up phase, venetoclax should be taken in the morning, to facilitate the necessary laboratory monitoring [123,124]. 

Venetoclax was initially used as continuous monotherapy until progression, showing high efficacy with an ORR of about 80% in RR-CLL, and achieving CR in a significant minority of patients (6–20%) [125]. Subsequently, fixed-duration combination treatment with venetoclax and anti-CD20 MoAb (Rituximab or Obinutuzumab) has been shown to improve overall quality and duration of response [10,11].

Venetoclax, whether as monotherapy or in combination therapy with an anti-CD20 MoAb, is generally well tolerated and manageable, with a substantially good safety profile, in both younger and older CLL patients with comorbidities. The most common AEs during venetoclax treatment should be carefully monitored, including the risk of TLS, hematological toxicities (with particular attention to neutropenia), and gastrointestinal symptoms.

### 3.1. Tumor Lysis Syndrome

Tumor Lysis Syndrome results from massive and rapid lysis of tumor cells. It can be classified on the basis of the intensity of the manifestations as either laboratory or clinical TLS [126,127].

In clinical trials where prophylactic measures to prevent TLS have been implemented, the incidence of TLS is very low, approximately 1.1–3.8%, with no substantial cases exhibiting clinical manifestations. However, outside of clinical trials, in real-world settings, the incidence of both laboratory and clinical TLS appears more frequent, reported at 5.7–13% and 2.7–6%, respectively [127].

Laboratory TLS is defined by the appearance of hyperuricemia, hyperphosphatemia, hyperkalemia, and hypocalcemia, which occur 6–24 h after the start of treatment, in the absence of any clinically significant manifestations. In contrast, clinical TLS is characterized by clinically relevant manifestations, such as renal insufficiency, cardiac toxicities (e.g., dysrhythmia), or neuromuscular symptoms (e.g., tetany, paresthesia, muscle twitching). It is induced by the worsening of the aforementioned metabolic abnormalities [126]. A practical management approach to TLS is summarized in Figure 3.

In the case of laboratory TLS, discontinue the next day’s dose. If laboratory abnormalities resolve within 24–48 h of the last dose, resume treatment at the same dose. If blood chemistry test alterations require more than 48 h to resolve, or in any clinical TLS event, resume treatment at a reduced dose after resolution of the event.

When venetoclax is used in combination therapy with obinutuzumab, the risk for TLS is higher after the first monoclonal antibody infusion. However, administering obinutuzumab before the venetoclax ramp-up reduces the CLL tumor burden, thereby reducing the TLS risk. Consequently, the TLS risk should be reassessed before the venetoclax ramp-up following obinutuzumab treatment.

### 3.2. Hematological Toxicity

The main hematological toxicities associated with venetoclax are neutropenia, thrombocytopenia, and anemia. These toxicities most frequently emerge during the five-week dose ramp-up phase and when venetoclax is combined with MoAbs, especially in the first three to six months of treatment. Rarely, severe cytopenias can appear subsequently. In the MURANO trial, RR-CLL patients were treated with the association of venetoclax with rituximab (VR) and reported grade 3 or 4 anemia or thrombocytopenia in 10.8% and in 5.7% of cases, respectively [11]. In TN-CLL patients, the combination of venetoclax and obinutuzumab (VO), as reported in the CLL14 trial, led to grade 3 or 4 anemia or thrombocytopenia in 8% and in 13.7% of cases, respectively [10]. Of relevance, the most common and significant hematological AE is neutropenia, which occurs more frequently when venetoclax is combined with anti-CD20 MoAbs. Higher rates of neutropenia are observed with obinutuzumab rather than rituximab, especially in the RR-CLL setting. The combination of venetoclax with MoAbs induces grade 3–4 neutropenia at rates of 57.7% for VR and 52.8% for VO, respectively. Neutropenia is more common in the first six months of combination treatment, compared to a rate of 11% when venetoclax is used in monotherapy. Despite this, rates of febrile neutropenia, at 3.6% VR and 6% VO, and grade 3–4 infections, at a rate of 17.5% for both VR and VO, are acceptable [10,11]. In the CLL13/GAIA trial, the combination of venetoclax with MoAbs induces grade 3–4 neutropenia at rates of 23.6% for VR and 16% and 22.8 for VO, respectively [129]. In real-world settings, venetoclax, alone or in combination with MoAbs, is well tolerated, with grade 3–4 neutropenia occurring in 62% of patients and neutropenic fever or infections occurring infrequently (2% of cases) [130].

#### Neutropenia Management

The management of neutropenia involves dose interruption, dose reduction, and growth factor support.

In the case of grade 3 neutropenia accompanied by infection or fever, or grade 4 neutropenia, venetoclax should be interrupted at first occurrence and then resumed at the same dose prior to the interruption. Granulocyte colony-stimulating factor (G-CSF) can be administered. If neutropenia recurs, venetoclax should again be interrupted and then resumed at a reduced dose (e.g., from 400 mg to 300 mg). G-CSF can be utilized in these instances as well. Prolonged cytopenia, despite the interruption or reduction of venetoclax, or the onset of cytopenia after six months of therapy, is relatively rare. In such cases, a bone marrow assessment should be considered to evaluate the potential emergence of other hematological disorders.

### 3.3. Gastrointestinal Events

The rates of nausea reported in CCL patients treated with venetoclax vary depending on the modality and duration of treatment. For venetoclax monotherapy, the rates are 42% for any grade and 1% for grade ≥ 3. For VR, the rates are 21%/1%, and, for VO, the rates are 19%/0%. Vomiting is very rare, with any grade occurring in less than 1% of cases. These symptoms appear manageable and respond well to antiemetic therapy [10,11]. The prevalence of diarrhea is high, but it is almost always of low grade (1–2). The rates of diarrhea vary by treatment modality: 43%/3% for venetoclax monotherapy, 40%/3% for VR, and 28%/4% for VO [10,11].

#### Gastrointestinal Toxicities Management

In case of grade 3 or 4 gastrointestinal toxicity, at first occurrence, venetoclax should be interrupted and could be resumed when the event improves to grade ≤ 1 or baseline. At subsequent occurrence, venetoclax should be interrupted and could be restarted at one dose-level reduction (e.g., 400 mg reduced to 300 mg) upon improvement to grade ≤ 1 or baseline.

It is important to exclude other conditions (infections, inflammatory bowel diseases, microscopic colitis, other drugs) as the cause of gastrointestinal symptoms before attributing them to venetoclax toxicity.

## 4. Class Effect Profile and Adverse Events Management: PI3kδ Inhibitors

The PI3K signaling pathway is integral to the pathogenesis of various neoplasms, including B-cell malignancies, with a notable impact on CLL. In CLL, PI3K inhibitors (PI3Ki) disrupt the B-cell receptor signaling pathway, affecting cell adhesion and leukemic cell trafficking, thereby inducing apoptosis and cell death [131]. Idelalisib, a selective inhibitor of the PI3K p110δ isoform, was the first PI3Ki approved by the FDA and EMA for CLL treatment. This approval was based on a double-blind, placebo-controlled, randomized phase III study that compared idelalisib plus rituximab with placebo plus rituximab in a heavily pre-treated RR-CLL population [132]. A second PI3Ki, duvelisib, received FDA approval in 2018 and EMA approval in 2021 for RR-CLL, based on the phase III DUO study, which demonstrated the superiority of duvelisib over the anti-CD20 agent ofatumumab in this patient population [133]. Other PI3Ki compounds have been considered for the treatment of CLL and other hematological and non-hematological malignancies [134]. Nevertheless, despite their proven efficacy, the role of PI3Kis in the treatment of CLL and B-cell malignancies was progressively downsized. This is due to the severe toxicity profile of this class of compounds and the increasing availability of highly effective and more tolerable agents, such as BTKis and BCL-2is. Furthermore, recent interim results of randomized trials involving PI3Ki agents have shown an alarming trend of a decrease in overall survival, with fatal and severe AEs in PI3Ki-treated groups compared to control arms. Consequently, idelalisib was voluntarily withdrawn from the market in 2022 by its developer in the US. Moreover, the class of PI3Kis underwent scrutiny by an FDA expert panel, which concluded that future FDA approval of PI3Kis should be supported by randomized data. This has led to the halting of clinical study development and substantial discontinuation of their approval in hematological malignancies [135]. Nevertheless, due to its clinical efficacy and mechanism of action, idelalisib is still used, albeit in a very small number of cases, in the treatment of CLL refractory to both covalent and non-covalent BTKis and BCL2is, according to ESMO guidelines [12]. This confers a residual but still significant role in the CLL treatment.

The most frequent and important toxicities of idelalisib are well known and substantially related to on-target, class-specific AEs, that include opportunistic infections—particularly Pneumocystis jirovecii pneumonia (PJP) and cytomegalovirus (CMV) reactivation—as well as autoimmune toxicities such as colitis, hepatitis, pneumonitis and cutaneous rash. The principal toxic effects of idelalisib and their clinical management have been fully described in different reports [136,137,138] and will not be detailed in this paper.

## 5. Combination Therapies: Multiple Drugs with More Adverse Events?

The rationale for combining BTKis and BCL-2is is based on several considerations. These TAs have distinct mechanisms of action, targeting two different and substantially independent cellular pathways both crucial for the biology of CLL cells. Of relevance, these two drug classes, beyond their distinct and complementary mechanism of action, also exhibit synergistic and additional effects. Venetoclax can sensitize CLL cells to BTKis, and ibrutinib, as well as the other BTKis, can inhibit the expression of two different BCL2-family antiapoptotic proteins, myeloid cell leukemia-1 (MCL-1) and B-cell lymphoma-extra large (BCL-XL), further sensitizing CLL cells to BCL-2 inhibition [139,140]. Moreover, the combination of both drug classes addresses different types of leukemic cells in various functional phases and sites within the organism, disrupting adhesion and homing, affecting the microenvironment supporting CLL cells in tissues, and inducing apoptosis in resting cells in the periphery [141,142].

Of great importance, the complementary and synergistic mechanism of action of BCL-2is and BTKis enhance each drug’s efficacy without increasing their toxicity. Essentially, the safety profile of the combination of ibrutinib and venetoclax is very similar to what could be expected by simply adding the specific side effects known for both compounds used in monotherapy. Several ongoing studies are exploring the efficacy and safety of the combination of ibrutinib with venetoclax (IV) in TN and in RR-CLL, with some differences in the schedule and durations among the studies, all considering a fixed duration treatment. Firstly, a single institution phase 2 study from MD Anderson Cancer Institute evaluated the efficacy and tolerability of IV association in a fixed duration approach for 24 cycles (28 days each cycle) in high-risk TN-CLL [143]. The phase 2/3 CAPTIVATE study considers an IV regimen in both an MRD-driven and fixed-duration approach in two different cohorts. In both cohorts, the same first 15 cycles are planned to start with ibrutinib alone for the first three cycles, adding venetoclax for the remaining 12 cycles, exploring efficacy and toxicity in CLL patients younger than 70 years [144,145]. The IV combination was also tested in the GLOW study, a randomized phase 3 study evaluating the IV regimen versus chlorambucil plus Obinutuzumab in older, unfit CLL patients, as defined by the Cumulative Illness Rating Scale (CIRS) [146]. Additionally, the FLAIR study, a phase 3 MRD-driven study, is exploring the efficacy of the IV combination, compared with the chemo-immunotherapy regimen FCR (fludarabine cyclophosphamide rituximab), in young, fit patients. In the IV arm, after two months of ibrutinib treatment, venetoclax is added for a maximum of six years, with the duration based on achieving an undetectable MRD in peripheral blood and bone marrow [147].

The efficacy and toxicity of the IV combination were also evaluated in RR-CLL in the CLARITY and HOVON141/VISION studies [148,149].

The safety profile of the combination of ibrutinib and venetoclax is very similar to what can be expected from the side effects known for ibrutinib and venetoclax when used as single agents. The most common AEs of the IV association in TN-CLL are grade 1–2 gastroenteric toxicities, including diarrhea (24–66%) and nausea (18–43%). Low-grade arthralgia is also frequent (30–48%). Neutropenia occurs in 20–40% of cases, with the majority being grade 3–4 (20–35%), similar to the neutropenia rate and severity expected for venetoclax used as monotherapy or in combination with anti-CD20 MoAbs. Atrial fibrillation occurs in 5–15% of patients, with a similar rate to what is reported for ibrutinib monotherapy, depending on the age and comorbidities of patients enrolled. Hypertension occurs in up to 15% of patients, with half of these cases being grade 3–4. Infection complications and febrile neutropenia are reported in up to 60% of patients, with a minority experiencing grade 3–4 infections. Of importance, this infection rate does not appear increased in the GLOW study with respect to the chlorambucil-obinutuzumab control arm and is similar to the infection rate observed in the venetoclax-obinutuzumab arm, as reported in the CLL-14 trial, in which the CLL patient population, considering age and comorbidities, are very similar to the GLOW trial [10,150].

Nevertheless, some cautions concerning cardiovascular complications could be derived from the GLOW study, in which the incidence of cardiac AEs, particularly AF and SCD, appear higher when comparing the IV regimen with trials using the same regimen on a younger and more fit population. The incidence of AF is 14.2% in the GLOW study versus 4% in the CAPTIVATE study. More importantly, seven (7%) treatment-related deaths occurred in the GLOW IV arm, including two cardiac and two sudden deaths, compared with no deaths due to cardiovascular complications in the chlorambucil-obinutuzumab arm. All these four patients had a CIRS score ≥10 and an ECOG score of 2. An update of the GLOW study after 55 months of follow-up was reported at the recent ASH 2023 Congress [151]. Interestingly, three further sudden deaths occurred after the end of treatment in the IV arm, and a further four cardiac and four sudden deaths occurred in the Chlorambucil-obinutuzumab control arm, indicating an intrinsic enhanced risk of cardiovascular incidents in this fragile CLL patient population. Considering the cardiac and cardiovascular toxicity of BTKis, the second-generation covalent BTKis acalabrutinib and zanubrutinib, which present a better safety profile, particularly regarding cardiovascular complication, could be considered preferable partners of venetoclax, coupling efficacy with better tolerance. Limited data have been reported concerning the association of acalabrutinib or zanubrutinib with venetoclax in a fixed duration treatment, explored in the phase III AMPLIFY and MAJIC trials for acalabrutinib or in the SEQUOIA-arm D (cohort with TP53 aberration) trial for zanubrutinib. Preliminary safety data with short follow-up suggest that these combinations are generally well tolerated, with no new particular safety signals identified [152].

Some ongoing studies are now evaluating the combination of a BTKi with venetoclax and an anti-CD20 MoAb. The triplet combination treatment appears to be a highly efficient treatment not yet evaluated and approved by national regulatory agencies and is not substantially used in the real-life treatment of CLL patients. From a safety point of view, these associations lead to an augmented rate of neutropenia and infections, as reported for the IVO arm of the CLL13/GAIA study. The effective advantage of introducing a MoAb as a third actor with a BTK and BCL2 inhibitor is under evaluation, and longer follow-up is needed [129,153].

Limited data are available from trials exploring the impact of second-generation BTKis combined with venetoclax and anti-CD20 MoAbs, as in the AVO (acalabrutinib, venetoclax, obinutuzumab) and ZVO (zanubrutinib, venetoclax, obinutuzumab) regimens, used as fixed duration or MRD-guided treatments [152,153,154,155,156]. High rates of ORR and CR have been reported for these trials. A low rate of discontinuations due to AEs has been reported. As expected, neutropenia and thrombocytopenia are the most frequent hematological AEs with AVO (75% all grade, 37% grade ≥ 3, and 73% all grade and 28% ≥ 3 respectively). AF occurs in only 3% of cases during the AVO regimen, and hypertension in 27%.

## 6. Drug–Drug and Drug–Food Interactions

TAs interact with various drugs and foods, potentially altering drug effectiveness or increasing toxicity. At the pharmacokinetic level, DDIs occur through modulation of the hepatic and intestinal CYP450 enzyme system by inhibition or induction, affecting drug bioavailability and exposure. Inhibition decreases pre-systemic metabolism, thereby increasing bioavailability and exposure to the drug and its side effects. Conversely, induction increases pre-systemic metabolism, leading to sub-therapeutic concentrations and reduced drug exposure [157]. DDIs may also occur at the pharmacodynamic level, when the pharmacological effect of one drug is modified by another, and can be categorized as synergistic, additive, or antagonistic [158].

Practical management strategies are crucial given the fact that the prevalence of polypharmacy is notably high among CLL patients receiving TAs, primarily owing to the management of concurrent health conditions in these aging patients. Regarding this matter, data from a systematic pharmacovigilance analysis among CLL patients who received ibrutinib in clinical practice reported that approximately two-thirds of patients were concurrently taking medications at the start of ibrutinib treatment, with a potential influence on drug metabolism [159].

The predominant pathway for the metabolism and clearance of all BTKis involves cytochrome P450 3A4 (CYP3A4), which contributes to the metabolism of numerous medications. Indications of BTKi management in case of concurrent administration with CYP3A4 inducers or inhibitors are reported in Table 3 [71,72,74,159,160,161]. 

Generally, strong inhibitors should be avoided with all BTKis. If co-administration with a strong CYP3A inhibitor becomes necessary, zanubrutinib can be used with an adjusted dose of 80 mg once daily. In the presence of moderate CYP3A4 inhibitors, the recommended dose of BTKIs should be reduced: for ibrutinib to 280 mg/day, for acalabrutinib to 100 mg once daily, and for zanubrutinib to 80 mg twice daily. There is no need for a dosage reduction when co-administering mild CYP3A4 inhibitors [160]. Strong inducers of CYP3A4 should be avoided. If co-administration becomes essential, acalabrutinib can be administered at a raised dose of 200 mg twice daily.

When an interacting medication is necessary for a short duration period (up to 7 days), a temporary interruption of the BTKi is an option. After discontinuation of the concomitant drug, it is recommended to restart the BTKi with the previous dose.

Venetoclax is metabolized by CYP3A4, and concurrent use with strong or moderate CYP3A4 inhibitors increases venetoclax exposure. This elevated exposure can heighten the risk of TLS during the ramp-up phase and increase the likelihood of other toxicities. Therefore, strong CYP3A4 inhibitors are contraindicated during the ramp-up phase. However, after the completion of the ramp-up phase, concomitant administration of strong inhibitors is permissible with a reduced venetoclax dose of 100 mg/day.

For moderate inhibitors, the venetoclax dose should be reduced by at least 50% at all phases of treatment. However, during the ramp-up phase, it is advised to avoid the administration of moderate inhibitors [124]. It is noteworthy that, in real-world contexts, co-administration with CYP3A4 inhibitors and polypharmacy does not affect the rate of persistent venetoclax dose reductions and definitive discontinuations due to toxicity; furthermore, this condition does not affect any survival outcome [162].

Co-administration with antifungals is particularly challenging with both BTKis and BCL2 inhibitors. If low doses of posaconazole suspension (≤400 mg/day) or voriconazole are concurrently administered with ibrutinib, the manufacturer suggests a dose reduction to 140 mg/day. However, when co-administered with higher doses of posaconazole suspension or alternative posaconazole formulations, the recommended dose is 70 mg/day [71]. Although there are no documented interactions between acalabrutinib and voriconazole or posaconazole, it is important to note that these azole agents are generally classified as moderate to strong CYP3A inhibitors.

In the case of venetoclax therapy and concomitant antifungal drugs, such as posaconazole or voriconazole, the anti-bcl2 dose is reduced by at least 75% and by 50% if co-administered with fluconazole or isavuconazole [163]. 

In terms of drug absorption, it is noteworthy that concurrent use of acalabrutinib with gastric acid-reducing agents can decrease acalabrutinib plasma levels. When such agents are necessary, it is advisable to prioritize histamine-2 receptor antagonists (H2-Ras) or antacids over proton pump inhibitors (PPIs), because the prolonged effects of PPIs may not be overcome by separating the times of administration. The intake of antacids should be spaced by at least 2 h from acalabrutinib [72]. Co-administration of zanubrutinib and ibrutinib with these gastric acid-reducing agents has not shown significant pharmacokinetic impacts.

The majority of BTKi trials did not allow the simultaneous use of vitamin K antagonists, and, as a result, in the clinical setting, it is not recommended. If there is a need for concurrent administration of oral anticoagulation and BTKis, clinicians should be mindful that apixaban and rivaroxaban undergo metabolism mediated by CYP3A4.

It is essential for patients and healthcare providers to understand the impact of certain foods on drug absorption and metabolism, as it can potentially result in significant clinical implications. Patients who consume grapefruit juice and other foods interacting with the CYP3A enzyme system may need to modify their diet accordingly. Furthermore, patients should be provided with clear guidance to promptly disclose the use of any over-the-counter products and supplements.

A summary of information useful for patients and caregivers is provided in Figure 4.

## 7. Infections

In the context of CLL treatment with current available therapies, managing the risk of infections is a critical aspect of patient care. Patients with CLL are inherently susceptible to infectious morbidity due to immunodeficiency associated with the disease. However, CLL-directed treatments can profoundly influence susceptibility to infections. This influence has been well established in the context of chemoimmunotherapy, but targeted agents are associated with an increased risk of infections [164,165,166].

The use of idelalisib significantly affects mature T-cells, neutrophils, and macrophages, leading to a high rate of serious AEs and increased mortality. These adverse events are mainly linked to infections, including PJP and CMV-related diseases [138,167,168]. 

The use of BTKis is associated with reductions in normal B-cells and antibodies, along with dysfunction in T-cells and NK-cells, and impaired phagocytic function in macrophages [169,170]. This immunological impairment leads to an increased risk of infections, including opportunistic infections [171,172], especially in the first months after treatment initiation [52]. Increased susceptibility to fungal infection has been reported during ibrutinib treatment [173,174,175]. However, the incidence of severe invasive fungal infections is relatively low, ranging from 1.2% to 3% of treated patients [176,177,178,179], and mainly occurs in heavily pretreated patients or those receiving concurrent steroid treatment [173,177,178]. Furthermore, treatment with single-agent ibrutinib can increase susceptibility to PJP, even in TN patients [180,181]. 

The occurrence of PJP in patients not receiving prophylaxis treated while receiving BTKi monotherapy is relatively low, with rates ranging from 2.4% to 5.2% [180,182]. In most cases, PJP was successfully treated and BTKi therapy resumed without issue [182]. Next-generation BTKis, such as acalabrutinib, zanubrutinib, and pirtobrutinib, have different specificities, potentially leading to variations in their impact on the immune system [52,177,183]. However, the rates of grade ≥ 3 infections are similar across different BTKis, ranging from 18% to 27% [52,166,184].

Neutropenia is the primary immunosuppressive effect associated with venetoclax, yet it appears to cause less frequent infections than chemoimmunotherapy-related neutropenia [166]. In clinical trials, the rate of grade 3 or higher infections with venetoclax ranges from 16% to 25%, with higher rates observed in patients with high-risk CLL [125,185,186,187]. The addition of MoAbs does not seem to increase the risk of severe infection, both in the RR setting (with 17.5% of patients experiencing grade ≥ 3 infections in the MURANO trial) [185] and in TN setting, even with the use of obinutuzumab (with 17.5% of patients having grade ≥ 3 infections in the CLL14 trial and 10.5% and 13.2% in the CLL13 trial in the VR and VO arms, respectively) [10,129]. Furthermore, the incidence of infections is significantly lower after fixed-duration treatment cessation [188].

When combining BTKis with venetoclax, the risk of infection does not seem to be higher, with an incidence of major infections ranging from 9% to 17% with the fixed-duration combination of ibrutinib and venetoclax in the TN setting [144,145,150]. As expected, in the RR setting the risk is higher, ranging between 17% and 28%, with the same combination [148,149]. The addition of obinutuzumab in the triplet combination with ibrutinib and venetoclax seems to confer additional infectious risk in the TN setting, with the incidence of major infections ranging from 19.5% to 21% [158,165]. In preliminary data, the triple combination with the next-generation BTKis acalabrutinib and zanubrutinib seems to result in a lower incidence of major infections [129,153].

The COVID-19 (coronavirus disease 2019) pandemic has deeply emphasized the intrinsic immune dysfunction of patients with CLL, both those undergoing treatment and those not in need of therapy [101,189]. Mortality remains relevant, and a significant number of patients develop post-COVID conditions, even with the improvement of prophylactic and therapeutic measures against SARS-CoV-2 (severe acute respiratory syndrome coronavirus 2), as well as the emergence of milder variants [190].

Given the high risk of morbidity and mortality due to severe infections, even in the context of novel agents, methods to stratify patients for their risk and measures to mitigate infectious risks, such as vaccination and prophylaxis, are of particular interest.

The literature extensively outlines factors linked with an elevated infection risk [165,191,192,193,194]. However, current guidelines do not provide clear indications on patient stratification, either at diagnosis or before treatment initiation [12,195]. We proposed an easy-to-use scoring system to identify diagnosis patients at high risk of developing severe infections, based on disease stage, age, IGHV (immunoglobulin heavy chain variable region) mutational status, and hypogammaglobulinemia [196]. The CLL-TIM is a machine-learning model designed to identify patients at risk of infection within two years of a CLL diagnosis [197]. Additionally, Mauro and colleagues developed a scoring system to evaluate the risk of infection in CLL patients treated with ibrutinib and/or rituximab [198].

Regarding vaccination, it is crucial to consider that immunological impairment in CLL patients also affects humoral vaccine response and seroconversion. Treatment may further suppress the response to vaccines, emphasizing the importance of keeping the vaccine status updated, including revaccination if prior vaccinations are expected to be ineffective [166]. Pausing BTKi treatment in the context of the SARS-CoV-2 vaccine has been shown to enhance immune response and may be a strategy to consider in patients on continuous BTKi treatment with a well-controlled disease status [199,200].

All patients with CLL can benefit from vaccination and are expected to receive vaccines for influenza, conjugated pneumococcal vaccine, recombinant varicella-zoster vaccine, Haemophilus influenzae type B, and the hepatitis B vaccine in seronegative patients. Vaccination status should be reviewed after the initiation of treatment [166,201,202].

Concerning COVID-19, vaccination is strongly advised [203]. Depending on local epidemiology, tixagevimab/cilgavimab can be considered as pre-exposure prophylaxis [202,204].

Prophylactic measures vary for different novel agents in CLL treatment. In all treated patients with a history of severe infections and severe hypogammaglobulinemia (IgG level of <4 g/L) immunoglobulin replacement therapy should be considered [195,205].

Idelalisib necessitates anti-PJP prophylaxis with cotrimoxazole, and CMV-DNA monitoring is advised, with pre-emptive treatment for reactivation [206].

For BTKi treatment, PJP prophylaxis is not mandatory, but it should be considered for patients with additional risk factors, such as prolonged steroid use and profound lymphopenia [206]. However, some authors advocate its use for all patients, whether TN or RR [207].

Patients on single-agent venetoclax are not at increased risk of particular opportunistic infections, obviating the need for specific antimicrobial prophylaxis [208]. Conversely, those receiving anti-CD20-MoAbs face a higher risk, necessitating PJP prophylaxis for certain patients with additional risk factors (e.g., corticosteroids, lymphopenia) [206].

Antifungal prophylaxis is not required, but it may be considered for patients treated with BTKis in an RR setting and with added risk factors (e.g., corticosteroids, prolonged neutropenia, other immunosuppressive drugs) [209]. 

All patients with occult hepatitis B infection (OBI) treated with anti-CD20 MoAbs should receive antiviral prophylaxis [210,211]. Venetoclax poses a potential risk of hepatitis B virus (HBV) reactivation, though no cases have been reported to date; thus, routine prophylaxis is not recommended when used as a single agent [212].

While BTKi use increases the risk of HBV reactivation, routine prophylaxis lacks sufficient supporting data [94,213,214]. Resolved HBV infection requires regular monitoring and pre-emptive antivirals should be initiated upon HBV DNA level rise [215,216,217,218,219]. Patients with positive HBsAg or detectable HBV DNA in resolved infection should receive anti-HBV prophylactic treatment with nucleotide analogs [219]. 

Regarding herpesvirus reactivation, data on venetoclax are insufficient to determine a specific risk for varicella-zoster virus (VZV) or herpes simplex virus (HSV) reactivation [220]. Some case reports showed the potential for fatal disseminated varicella zoster in patients treated with BTKis [221,222,223], while safety analysis from clinical trials of ibrutinib showed up to 5% of patients with herpes zoster or oral herpes disease [32]. Therefore, there is not sufficient data to generally recommend antiviral prophylaxis in all patients treated with BTKis, whereas some authors advocate its use in ibrutinib-treated patients [54,207]. Awareness of potential risks is essential, and individualized antiviral prophylaxis might be considered [220]. With PI3Kδis, despite limited data on HSV and VZV reactivation rates [224], antiviral prophylaxis is widely used and recommended for all patients receiving idelalisib [220,225].

As per the management of TA, there is still no consensus during infectious events. Patients undergoing targeted therapies should undergo monitoring for fever, neutropenia, and infection. Additionally, a thorough investigation for etiology and early treatment should be initiated when an infection is documented [226].

The venetoclax brochure indicates holding the drug at the first occurrence of an infectious event of grade 3–4 until resolution or reduction to grade 1. If a second event occurs, the drug should be resumed at a lower dosage [123,124]. In addition, infections are one of the main causes of treatment discontinuation or drug dosing modifications [162]. However, there is no evidence that holding venetoclax has a positive impact on a specific infection outcome outside of the case of severe neutropenia.

With BTKis, no clear indication is given in labeled indications. In the event of severe infections (grade ≥ 3) or related complications, some suggest that ibrutinib treatment should be withheld and the causative agent(s) of infection determined. Upon infection resolution, ibrutinib should be reintroduced at an adequate dose [201,226]. Conversely, some experts do not recommend discontinuing treatment unless during grade 4 infections and until resolved to at least grade 3 [54]. This is even considering the risk of disease flare in case of BTKi discontinuation and the potential worsening of immune dysfunction with disease activity [227].

For the management of PI3Kδ see reference [138]. 

A summary of infection risk management with novel agents is provided in Figure 5.

## 8. Secondary Primary Malignancies 

The occurrence of second primary malignancies (SPMs) subsequent to a CLL diagnosis can affect morbidity and offset the increased longevity observed in CLL patients. Consequently, comprehending the nature and extent of this issue in CLL is crucial for effective health-related planning and surveillance activities [228].

Population-based studies indicate that CLL patients have a higher likelihood of developing secondary primary malignancies (SPMs) compared to the general population. In a 30-year follow-up study, Kumar et al. found cumulative SPM incidences of 29.45% at 10 years, 50.84% at 20 years, and 69.18% at 30 years. These SPMs were typically detected after a median follow-up of 4.2 years post-CLL diagnosis, with patients having a median age of 74 years [229]. Furthermore, the study indicated that treated CLL patients experienced a 42% higher occurrence of SPMs, particularly hematological malignancies, than the general population. There were no significant differences in SPM incidence across different first-line treatment regimens. In contrast, untreated CLL patients did not show an increased risk of SPMs compared to the general population [229]. Falchi et al. found that, among patients with more than 10 years of survival after CLL diagnosis at MD Anderson, SPMs were detected in 36% of cases, with an increased risk in males and patients of less than 60 years age [230]. In addition, a high rate of SPMs after frontline FCR-based therapies was observed and confirmed, particularly contributed to by t-AML/MDS [231,232,233,234].

In the last year, Chatzikonstantinou et al. presented results from a large retrospective international multicenter study on consecutive patients diagnosed with CLL/SLL between 2000 and 2016 at each participating site. Almost 16% of patients developed at least one other SPM following CLL diagnosis. Notably, these patients exhibited a lower OS, particularly those diagnosed with AML or MDS. Finally, data confirmed that the occurrence of AML or MDS was specifically associated with the administration of the FC ± R regimen [235].

Referring to targeted agents, the incidence of SPMs reported in clinical trials until now seems to be similar to that of comparative conventional cytotoxic chemotherapy agents. More specifically, the incidence of SPMs during BTKi treatment ranges from 11% to 19% depending on the line of treatment and follow-up, with more than 50% of tumors being non-melanoma skin cancers [30,38,102]. Similar rates of SMPs were found with venetoclax-based combinations and these rates were similar to those of chemoimmunotherapy comparator arms. Rates range from 8.9% to 12.7% for VO [129,236] and from 10.1% to 10.8% for the VR arm [11,129], with more than 50% of tumors being non-melanoma skin cancers in all cases. 

These findings point out the relevance of implementing routine surveillance and applying targeted interventions to mitigate the heightened morbidity and mortality linked to SPMs, thereby enhancing long-term outcomes for CLL patients. All treated patients with CLL/SLL should undergo age- and sex-appropriate screening measures for solid tumors.

## 9. Quality of Life (QoL), Cognitive Impairment, and Psychological Distress

A diagnosis of chronic leukemia and, in particular, of CLL, represents an emotional challenge for patients. Discussing the characteristics of the disease, including the long-term follow-up, the clinical aspects in case of progression, and emphasizing the expanding array of treatment options can alleviate the concerns of most patients. However, all the treatments, including the strategy of watchful waiting, can adversely affect cognitive and psychological health, leading to issues like depression and anxiety [237,238,239,240].

Chemotherapy and new target therapies have been associated with reduced cognitive performance in cancer patients [238]. Conversely, recent advances in treatment for CLL have shown promising results. For instance, a phase II study combining obinutuzumab, ibrutinib, and venetoclax demonstrated either preserved or improved cognitive performance, particularly benefiting those whose jobs demand mental sharpness or older adults at risk of treatment-induced cognitive decline [241]. Moreover, the combination of acalabrutinib and obinutuzumab has not only maintained an adequate QoL during treatment but, in some cases, has gradually induced an improvement, due to a better health status [242]. Similarly, patients treated with ibrutinib reported a higher QoL and enhanced social functioning compared to those receiving conventional immunochemotherapy [239,243,244,245]. 

The dynamic evolution of leukemia treatment is promising for patient outcomes, extending beyond mere overall survival and progression-free survival, to encompass the preservation and enhancement of QoL. 

Furthermore, the introduction of new therapies and combinations, along with ongoing research, emphasizes the significance of integrating psychological and cognitive support into patient treatment and care planning. This greatly reinforces the concept of a holistic approach to leukemia management.

## 10. Conclusions and Future Challenges

Our CLL patients, and we as physicians, can consider ourselves fortunate to live in an era in which such a powerful arsenal of new molecules and combinations is available. We are witnesses and participants in major changes in the treatment of CLL due to the introduction and common use of therapies able to achieve better control of the disease in terms of time to progression, time to next treatment, and overall survival.

Optimal management of AEs is crucial to achieving good outcomes and maintaining quality of life. Given the specific AE profiles of these therapies, the therapeutic choice extends beyond age and fitness to a holistic approach that considers specific comorbidities and patient needs. A multidisciplinary approach is essential to manage the complexities of CLL patients effectively. Comprehensive AE management should begin with selecting the appropriate compounds and treatment duration, taking patient comorbidities into account for better tolerance. AEs must be well-known, promptly identified, monitored, and managed to maximize efficacy and avoid inadequate dose reductions and treatment discontinuations that could compromise the effectiveness of the therapies.

Some AEs are more relevant and require special attention. For instance, BTKis, particularly ibrutinib, are predominantly associated with cardiovascular toxicity, including an increased risk of AF and hypertension. However, proper management of BTKi treatment also needs to address additional AEs that might be considered less relevant from a clinical point of view, such as bleeding, arthralgias, myalgias, hematological toxicities, and diarrhea. These further AEs are generally low-grade and self-limiting but can determine permanent discontinuation, impacting treatment efficacy. Optimal management of AEs is crucial to avoid discontinuation and maintain quality of life. Strategies include supportive care, dose reduction, or switching to second-generation covalent BTKis. Supportive care is essential both in managing cardiovascular toxicity and enhancing overall BTKi tolerability, addressing side effects like diarrhea, arthralgia, and musculoskeletal pain with appropriate medications. While dose reductions are suggested for managing AEs, whether this practice has a positive outcome on the risk of cardiac toxicity, bleeding, or immunization failure is still unproven. Initial clinical trials indicated that a higher dose intensity of ibrutinib was associated with better clinical outcomes [39]. However, real-world data showed that AE-related discontinuation led to worse survival, while persistent dose reductions did not adversely impact outcomes [246]. Additionally, recent real-world data and pooled analyses from clinical trials suggest that lowering the ibrutinib dose may reduce the risk of AE recurrence without compromising treatment efficacy [247,248]. This question remains largely unanswered, and further research is needed. Switching to second-generation BTKis is a valuable strategy. Acalabrutinib and Zanubrutinib are effective alternatives to ibrutinib, offering better tolerability with similar efficacy. 

With venetoclax-based fixed-duration treatment strategies, we can achieve profound responses with undetectable MRD, implying durable remission with limited AEs during the treatment period. Importantly, these results can also be obtained, with some limitations, in high-risk patients. Previously, profound responses could only be achieved with an intensive chemo-immunotherapy approach, such as the FCR protocol, at the cost of significant toxicity, both short- and long-term, including immunosuppression, cytopenia, infections, and second primary neoplasia. In addition to TLS and IRR, which can limit the initiation of venetoclax-based treatments, cytopenias—particularly neutropenia—are the most prevalent adverse events. Careful management of these AEs is essential to ensure treatment efficacy. Reducing the dosage may be a good strategy to limit the need for interruption and permanent discontinuation. 

Different trials are ongoing to evaluate the efficacy and safety of novel TA associations, with or without MoAb, including triple-drug associations, in CLL patients of various ages, comorbidities, and biological characteristics. Moreover, some novel and promising agents are emerging in the landscape of CLL treatment, including non-covalent BTKis, particularly pirtobrutinib, which is already FDA-approved and effective in covalent-BTKi-resistant CLL. Additionally, different novel BCL-2is and new classes of compounds targeting BTK, such as BTK degraders or dual covalent/non-covalent BTis, are being developed. Immunotherapies, including chimeric antigen receptor T-cells (CAR-T cells) and bispecific MoAbs, are also being evaluated. All these compounds and procedures are very promising and potentially capable of improving efficacy and disease control in CLL. These treatments will be more efficient if AEs and toxicities can be contained and reduced, especially considering the typical CLL patients, where aging and comorbidities significantly impact the disease and its treatment and must be taken into strong consideration.

## Figures and Tables

**Figure 1 cancers-16-01996-f001:**
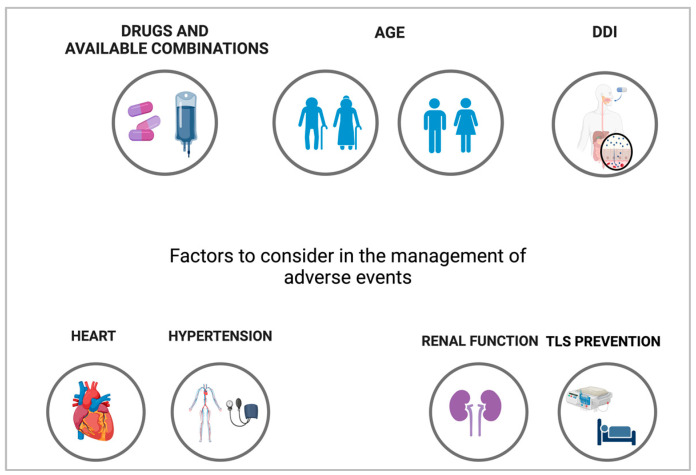
Main factors influencing treatment selections and toxicity development and management. DDI: drug–drug interaction; TLS: Tumor Lysis Syndrome.

**Figure 2 cancers-16-01996-f002:**
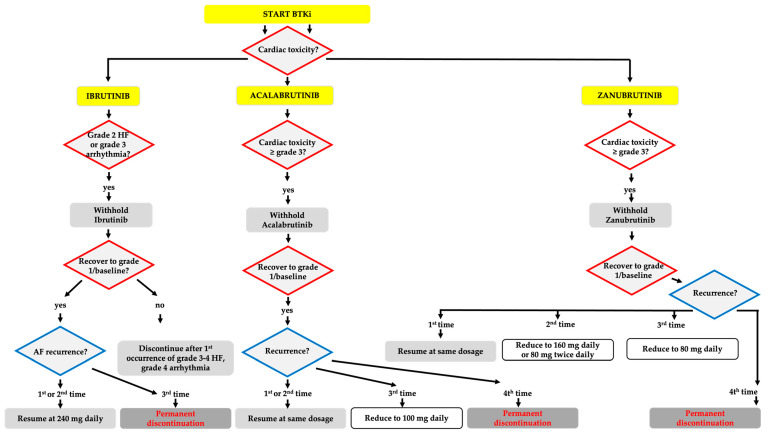
General management of BTKis in case of cardiac toxicity of grade 3–4 [70,71,72,73,74,75].

**Figure 3 cancers-16-01996-f003:**
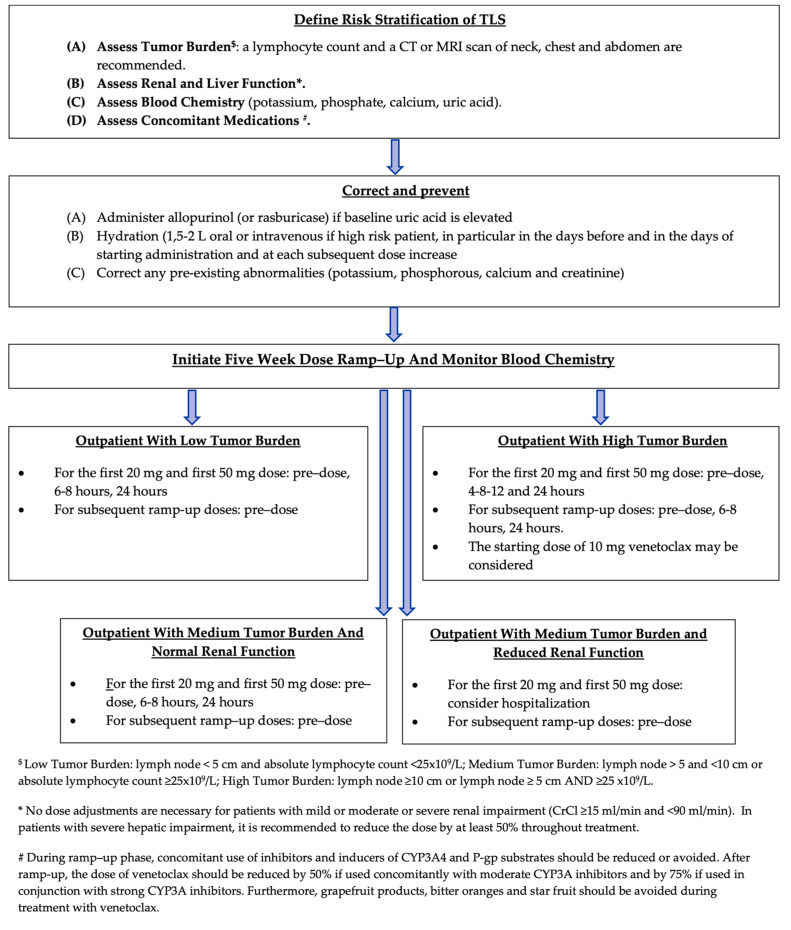
Management of TLS [11,122,123,124,128].

**Figure 4 cancers-16-01996-f004:**
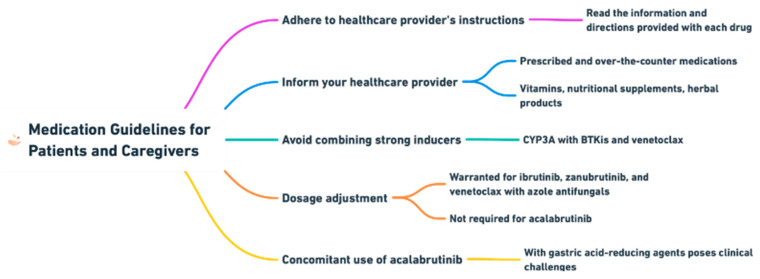
Summary of indications for patients and caregivers to avoid significant DDIs.

**Figure 5 cancers-16-01996-f005:**
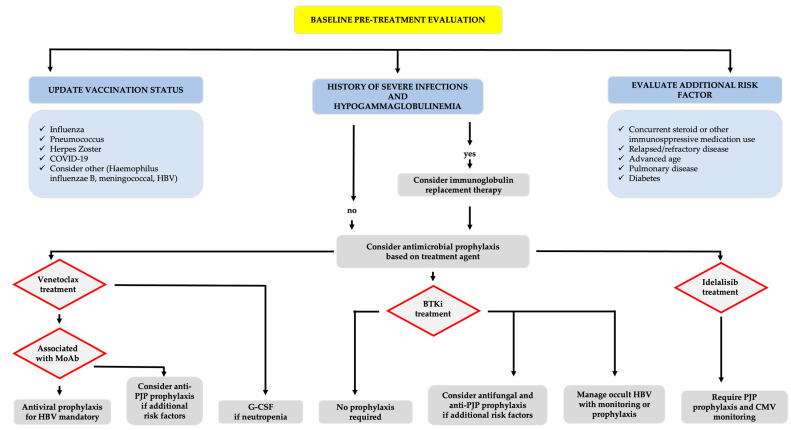
Management of infectious risk in patients with CLL undergoing treatment with TAs.

**Table 1 cancers-16-01996-t001:** Incidence, mechanisms and management recommendations for BTKi-associated cardiological AEs and bleeding.

Adverse Events	BTKi	Incidence (Any Grade/Grade ≥ 3)	Mechanism	Management
**Atrial** **Fibrillation**	Ibrutinib	16%/2–5%	TEC/HER2/HER4 strong inhibition	− **A**void stroke: anticoagulation − **B**etter symptom control: rate control vs. rhythm control − **C**ardiovascular and other comorbidities management
	Acalabrutinib	6–9% */1–5% **	HER4 strong inhibition	
	Zanubrutinib	3–6% ^^^/≤1%	HER4 strong inhibition TEC weak inhibition	
**Ventricular** **Arrhythmia**	Ibrutinib	0.3%/1%	Altered Calcium homeostasis in cardiomyocytes Myocardial fibrosis IGF-1pathway off target inhibition	− Treatment definitive discontinuation
	Acalabrutinib	0.4%/0.4%		
	Zanubrutinib	0.8%/0.8%		
**Hypertension**	Ibrutinib	16–23% ^§^/8–12% ^§§^	Off-target kinase inhibition? Systemic inflammatory changes?	− Correct predisposing factors − antihypertensive therapy
	Acalabrutinib	7–9% ^^^^/3–4% ^^^^^		
	Zanubrutinib	14–17% ^°^/6–15% ^°°^		
**Bleeding**	Ibrutinib	36–51%/3–4%	BTK/TEC strong inhibition	− Minor bleeding: no intervention − Major bleeding: ○ consider treatment discontinuation; ○ platelet transfusions regardless of platelet count.
	Acalabrutinib	36–51%/3%	BTK strong inhibition	
	Zanubrutinib	36–45%/3%	BTK strong inhibition	

* ELEVATE-TN: 6%; ELEVATE R-R: 9% ** ELEVATE-TN: 1%; ELEVATE R-R: 5% ^^^ SEQUOIA: 3%; ALPINE: 6% ^§^ RESONATE-2/ELEVATE R-R: 23%; ALPINE: 16% ^§§^ RESONATE-2: 8%; ALPINE: 16% ^^^^ ELEVATE-TN: 7%; ELEVATE R-R: 9% ^^^^^ ELEVATE-TN: 3%; ELEVATE R-R: 4% ^°^ SEQUOIA: 14%; ALPINE: 17% ^°°^ SEQUOIA: 6%; ALPINE: 15%. TEC: tyrosine kinase expressed in hepatocellular carcinoma; HER2: human epidermal growth factor receptor 2; HER4: human epidermal growth factor receptor 4; IGF-1: insulin-like growth factor 1.

**Table 2 cancers-16-01996-t002:** Incidence, mechanisms, and management recommendations for BTKi-associated selected AEs.

Adverse Events	BTKi	Incidence (Any Grade/ Grade ≥ 3)	Mechanism	Management
Neutropenia	Ibrutinib	25–39%/13–31%	On-target toxicity	Growth factor support
	Acalabrutinib	21–23%/13–19%		
	Zanubrutinib	37–44%/15–19%		
Thrombocytopenia	Ibrutinib	13–29%/3–8%	On-target toxicity	− Dose modifications or withholding in severe cytopenias − Transfusion threshold of 25,000/uL − Bone marrow evaluation to rule out other causes
	Acalabrutinib	15–32%/3–10%		
	Zanubrutinib	22–27%/<1–4%		
Diarrhea	Ibrutinib	22–59%/<1–4%	EGFR inhibition	− Symptomatic treatment and dose adjustments − Dietary modifications, hydration, anti-diarrheal medications − Probiotics
	Acalabrutinib	18–39%/1–5%		
	Zanubrutinib	14–18%/<1–2%		
Dermatological toxicity, Nail and Hair Changes	Ibrutinib	21–24%/<1–3%	EGFR inhibition	− Topical corticosteroid therapy − Oral antihistamines and systemic corticosteroids for severe rash − Temporary interruption of treatment
	Acalabrutinib	9–25%/<1%		
	Zanubrutinib	20–28%/1%		
Arthralgias and Myalgias	Ibrutinib	17–28%/<1%	Likely related to off-target inhibition	− NSAIDs, acetaminophen, or corticosteroids with caution due to bleeding risk
	Acalabrutinib	15–32%/1%		
	Zanubrutinib	26–38%/1–3%		
Headaches	Ibrutinib	14–18%/1–2%	Off-target effect Possibly CGRP agonism	− Moderate dose of caffeine or acetaminophen
	Acalabrutinib	22–39%/<1%		
	Zanubrutinib	11–12%/0–1%		

EGFR: epidermal growth factor receptor; NSAIDs: non-steroidal anti-inflammatory drugs; CGRP: calcitonin gene-related peptide.

**Table 3 cancers-16-01996-t003:** Management recommendations for BTKis with CYP3A modulators and selected interactors.

Interacting Agents	Ibrutinib	Acalabrutinib	Zanubrutinib
Strong CYP3A inhibitors	avoid	avoid	80 mg OD
Moderate CYP3A inhibitors	280 mg OD 140 mg OD with voriconazole 70 mg OD with posaconazole	100 mg OD	80 mg BID
Strong CYP3A inducers	avoid	avoid	avoid
PPI	-	avoid	-
Grapefruit, St John’s wort, Seville Oranges	avoid	avoid	80 mg BID
Warfarin/ Vit K antagonists	avoid	avoid	avoid

OD: once daily; BID: bis in die; PPI: proton pump inhibitors.

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
