# Peer review of "Chronic Lymphocytic Leukemia: Management of Adverse Events in the Era of Targeted Agents"

_cancers, 2024, doi:10.3390/cancers16111996_

Round 1

Reviewer 1 Report

Comments and Suggestions for Authors

Andrea, et al report a comprehensive manuscript about the adverse effects and management of TKi in CLL. It is very important. However, some points need to be clarified.

1. Line 55, does MoAb mean CD20 MoAb?

2. Line 56-57, the citated reference [9] is review article. Please also cite original article.

3. There are many abbreviations without full name when they appear, eg, figure 1, “DDI”, line 248, “SCD”, line 360, 361, “GPVI”, “P2Y12”, etc

4. in 2.1.2 paragraph, please mention the possible mechanisms of VA.

5. Figure 1 is not in good resolution. Please redraw and make it clear.

overall, it is a good manuscript.

Author Response

We thank Reviewer 1 for the valuable comments, which have significantly contributed to improving our manuscript. Below are our responses to each point raised:

  1. The text has been updated to specify "CD20," making it clear that MoAb refers to CD20 monoclonal antibodies
  2. The original research articles have been added alongside the review article to provide comprehensive references
  3. All abbreviations have been reviewed and defined upon their first appearance throughout the manuscript, including in Figure 1.
  4. A detailed explanation of the possible mechanisms of ventricular arrhythmias (VA) associated with BTK inhibitors has been added to paragraph 2.1.2. : “The factors contributing to VAs associated with BTKi remain largely unidentified. The mechanisms underlying VAs induced by BTKi are still not fully understood. However, disruptions in calcium handling and myocardial fibrosis, altered repolarization dynamics, and direct effects on cardiac ion channels have been described as associated with BTKi treatment, leading to an increased risk of arrhythmic ventricular events [63,64].” Table 1 has been updated accordingly.
  5. Figure 1 has been redrawn and a high-resolution version has been uploaded to enhance clarity and readability.

Reviewer 2 Report

Comments and Suggestions for Authors

This is a very complete and good review on managing the toxicities associated with the new targeted therapies. The review was well referenced and included recent studies. 

1. While suggestions were made regarding dose reductions with myelosuppression and other side effects, which is done commonly in the "Real-World" setting. However, does a reduced dose of the BTKis also decrease the risk of atrial fibrillation/hypertension, bleeding or response to immunization? It likely is not known, and this should be stated.

2. There are quite a few grammatical errors, particularly after the venetoclax section, which should be fixed. The review is also wordy and would be improved by shortening and having less repetition. Some of the writing is also difficult to follow, eg, the sentence starting on Line 636 is very convoluted and needs to be rewritten.

3, Sections 10 and 11 should be combined and markedly shortened, as the

Comments on the Quality of English Language

/

Author Response

We thank Reviewer 2 for the valuable comments, which have significantly contributed to improving the overall quality of our manuscript. Below are our responses to each point raised:

  1. We have added the following statemnt to the text: "While dose reductions are suggested for managing AEs, whether this practice has a positive outcome on the risk of cardiac toxicity, bleeding, or immunization failure is still unproven. Initial clinical trials indicated that higher dose intensity of ibrutinib was as-sociated with better clinical outcomes [39]. However, real-world data showed that AE-related discontinuation led to worse survival, while persistent dose reductions did not adversely impact outcomes [246]. Additionally, recent real-world data and pooled analyses from clinical trials suggest that lowering the ibrutinib dose may reduce the risk of AE recurrence without compromising treatment efficacy [247,248]. This question remains largely unanswered, and further research is needed."
  2. We have revised the text for grammar, better clarity, and fluency. The sentence starting on Line 636 (now 644) has been rewritten for better readability.
  3. We have combined and shortened sections 10 and 11 accordingly, also as per Reviewer 4’s comments

Reviewer 3 Report

Comments and Suggestions for Authors

Very good written. Very good summary of side effect, the potential causes of them and possible diet and drug treatment of them.

Author Response

We thank Reviewer 3 for the appreciation of our work. We appreciate the positive feedback and are glad that the manuscript was well-received.

Reviewer 4 Report

Comments and Suggestions for Authors

The authors reviewed extensively adverse events of targeted agents and their managements in the current treatments of CLL. Although this review article is highly informative for hematologist and physicians, the manuscript preparation has some problems.

Major comments:

1.       The paragraph 10 (Hot spots in managing adverse events) should be deleted because the content of this paragraph is only the repeating of previous descriptions.

2.       In accordance to Table 2, Section 2.5, Headaches, section 2.6., and section 2.7 should be section 2.7., 2.5., and 2.6.

Minor comments:

1.       Although there are many abbreviations properly used, still many abbreviations to be defined first with full term; for example, BTKi (line 45), BCL-2 (line 45), AEs  (line 57), CYP3A (line 65 ), SLL (line 82), ITK, TEC, EGFR, CSK, BMX, BLK, FGR, and JAK3 (lines 94-95), MCL (line 125), R/R (line 141), PI3K and AKT (line 169), CMR (line 321), BH3 (line 480), G-CSF (line 563), MCL-1 and BCL-XL (line 630), FCR (line 655), COVID-19 (line 845), SARS-Cov-2 (line 849), IGHV (line 856), CMV (line 879), HBV (897), and VZV and HSV (line 903).

2.       Conversely, once abbreviated, please use only the abbreviation but not full term writing. There are many such confusions in this manuscript. Please correct them throughout the text.

3.       Line 486; Venetoclax (Ven): Abbreviation is not needed in this passage.

4.       Figure 1: Please define DDI and TLS in the legend.

5.       Table 1; Title; BTKi associated →BTKi-associated

Please define TEC and HER2, and HER4 in the legend. Please avoid italics.

6.       Figure 2: What is cBTKI?

7.       Table 2; Please define EGFR, NSAIDs, and CGRP in the legend. Gastrointestinal toxicity should be changed as Diarrhea. Dermatological toxicity should be changed as Dermatological toxicity, Nail and Hair Changes.

8.       Figure 3; The letter size of legends is too small.

9.       Figure 4; The letter size is too small.

10.    Figure 5; Please avoid italics. 

11.    Tyrosine Kinese, etc. should be written as tyrosine kinase unless the word is a proper noun throughout the manuscript including legends for Table 3.

12.    Line 348: Is TKi correct?

13.    Line 32: evaluating may be better than examining.

14.    Line 1074: TA AE should be as AE caused by TAs.

Comments on the Quality of English Language

Generally, English writing is good.

Author Response

We thank Reviewer 4 for the valuable comments, which have significantly contributed to improving our manuscript. Below are our responses to each point raised:

Major comments

  1. We have merged paragraph 10 with paragraph 11 and markedly shortened it to avoid repetition, as also suggested by Reviewer 3.
  2. We have revised the sections accordingly for better alignment with Table 2

Minor comments:

  1. We have clarified the full names of all abbreviations upon their first appearance throughout the manuscript, including those listed.
  2. We have ensured that abbreviations are used consistently throughout the text after their first definition.
  3. We have removed the abbreviation "Ven" from this passage
  4. We have added definitions for DDI (drug-drug interaction) and TLS (tumor lysis syndrome) in the legend of Figure 1.
  5. We have revised the title to "BTKi-associated" and defined TEC, HER2, and HER4 in the legend. Italics have been removed.
  6. We have changed cBTKi (covalent BTKi) to the easiest form BTKi in the figure for better clarity.
  7. We have defined EGFR, NSAIDs, and CGRP in the legend and made the suggested changes to the table.
  8. We have increased the letter size of the legends in Figure 3.
  9. We have increased the letter size in Figure 4.
  10. We have removed italics from Figure 5.
  11. We have corrected all instances to lower case throughout the manuscript and in the legends for Table 3.
  12. It was intended for tyrosine kynase inhibitor, we wrote it as full name for better clarity.
  13. We have changed "examining" to "evaluating".
  14. We have revised "TA AE" to "AE caused by TAs."

Reviewer 5 Report

Comments and Suggestions for Authors

This review covers the management of adverse events from newer, targeted agents for the treatment of CLL. Although this is well-covered ground, this review is admirably comprehensive and may be a "one stop shop" for the subject. Figures are helpful and generally attractive. References are extensive. Overall, this review is expected to be useful for clinicians managing adverse events caused by targeted agents against CLL. 

Comments on the Quality of English Language

No significant issues

Author Response

We thank Reviewer 5 for the valuable comments and the appreciation of our work. We appreciate the positive feedback and are pleased that the manuscript is considered comprehensive and useful.